# To what extent does confounding explain the association between breastfeeding duration and cognitive development up to age 14? Findings from the UK Millennium Cohort Study

**Reneé Pereyra-Elías** [ID]*, **Maria A. Quigley, Claire Carson**

Nuffield Department of Population Health, National Perinatal Epidemiology Unit, University of Oxford, Oxford, United Kingdom

* renee.pereyraelias@balliol.ox.ac.uk

**Data Availability Statement:** The datasets used in this article are publicly available in the UK Data Service at http://doi.org/10.5255/UKDA-SN-8172-

## Abstract

### Background

Breastfeeding duration is associated with improved cognitive development in children, but it is unclear whether this is a causal relationship or due to confounding. This study evaluates whether the observed association is explained by socioeconomic position (SEP) and maternal cognitive ability.

### Methods

Data from 7,855 singletons born in 2000–2002 and followed up to age 14 years within the UK Millennium Cohort Study were analysed. Mothers reported breastfeeding duration, and children's cognitive abilities were assessed at 5, 7, 11, and 14 years using validated measures. Standardised verbal (age 5 to 14) and spatial (age 5 to 11) cognitive scores were compared across breastfeeding duration groups using multivariable linear mixed-effects models (repeated outcome measures).

### Results

At all ages, longer breastfeeding durations were associated with higher cognitive scores after accounting for the child's own characteristics. Adjustment for SEP approximately halved the effect sizes. Further adjustment for maternal cognitive scores removed the remaining associations at age 5, but not at ages 7, 11 and 14 (e.g.: verbal scores, age 14; breastfed ≥12 months vs never breastfed: 0.26 SD; 95%CI: 0.18, 0.34).

### Conclusion

The associations between breastfeeding duration and cognitive scores persist after adjusting for SEP and maternal cognitive ability, however the effect was modest.

4. Data citation: University of London, Institute of Education, Centre for Longitudinal Studies. Millennium Cohort Study: Longitudinal Family File, 2001-2018. [data collection]. 4th Edition. UK Data Service, 2020 [Accessed 16 February 2021]. Available from: http://doi.org/10.5255/UKDA-SN-8172-4.

**Funding:** This work was supported by Nuffield Department of Population Health at the University of Oxford, as part of a DPhil Scholarship held by RPE.

**Competing interests:** The authors have declared that no competing interests exist.

## Introduction

The association between breastfeeding and cognitive development has been extensively investigated. A systematic review found that on average breastfed infants scored 3.44 points higher in standardised intelligence tests than their non-breastfed peers [1], however, a causal relationship is still debated. It is argued that improved cognitive outcomes could be explained by other characteristics of the women who breastfeed their babies, principally socioeconomic position (SEP) and maternal intelligence [1–6].

In developed economies, including the UK, women from more socioeconomically advantaged backgrounds are more likely to breastfeed their infants [7], and to breastfeed for longer [8]. SEP refers to a range of factors that influence the position of individuals within society, and is reflected by education, occupation, wealth, and income, among others, which usually shape the distribution of health, social and wellbeing outcomes [9]. In this case, a higher SEP may be associated with a more favourable nurturing environment, less adversity, and more opportunities for intellectual stimulation, all of which may influence child cognitive outcomes [10]. Moreover, maternal intelligence also tends to be associated with longer breastfeeding durations [1–3] and is an important predictor of intelligence in the offspring [11]. Despite the potential influence of these variables on the association of interest, systematic reviews have shown that many studies do not conduct sufficient adjustment for potential confounders, with maternal intelligence being one of the most frequently overlooked variables [1–2]. For example, in the systematic review by Horta et al [1], only nine of the 17 studies included had adjusted for maternal cognitive ability. Additionally, the majority of studies that evaluate cognitive outcomes do so at young ages only (first years of life) and have relatively limited sample sizes [1–3].

Most of the studies only controlling for SEP find that longer breastfeeding durations are associated with higher cognitive scores [12–19]. However, others report no association after adjusting for SEP [20,21]. Given that the association between SEP and breastfeeding can vary by setting (no or inverse association in low- and middle-income countries) [22], one of the proposed ways to build the case for causal inference is to replicate this research question in populations with different confounding structures [4]. Results from a Brazilian sample, where breastfeeding patterns did not vary considerably with SEP, showed that the association persisted after adjustment for socioeconomic circumstances [5]. Additional adjustment for maternal intelligence removes the observed associations in some studies [23–25], but not in others [26–31]. The only experimental study conducted to date randomised a breastfeeding promotion intervention (rather than breastfeeding duration *per se*)—which by design would be controlling for SEP, maternal intelligence, and other confounders—and found a positive association between breastfeeding duration and child cognitive abilities at age 6 [32].

The aim of the present study was to evaluate how much of the association between breastfeeding duration and cognitive development is due to confounding by SEP and maternal cognitive scores among children from the UK Millennium Cohort Study (MCS). While this association has been explored previously in the MCS, there are gaps yet to be filled. Previous studies have only evaluated cognitive outcomes up to age 7 [16,33–36], while data on cognitive development is currently available at ages 11 and 14. Additionally, given that the maternal cognitive measures have been collected only recently, none of the previous studies have controlled for their effect. Therefore, the associations found on these studies [16,33–36] may be attributable to an important source of residual confounding.

## Methods

### Study population

The MCS is a nationally representative cohort study that recruited 18818 children aged around 9 months living in the UK in 2000–2002 and followed them up at ages 3, 5, 7, 11, 14 and 17

years [37] (**Fig 1**). Our study sample included singleton births with a gestational age at birth ≥37 weeks. Multiple birth and length of gestation can influence both breastfeeding patterns [38], and cognitive outcomes [39–41] and thus could introduce substantial confounding, which was controlled by restriction. Additionally, it is possible that the effect of BF duration on cognitive outcomes might be different among children who were twins/multiples and those born premature [16,42]. Those participants for whom the mother was not the main survey respondent and those whose mothers did not speak English were also excluded.

Given that this is a longitudinal study in which the outcome was measured on multiple occasions, children were only excluded if they were missing all data for the outcome measures. Further exclusions were made based on missing data for the exposure, and covariates, and were specific for the analysis of each cognitive outcome.

## Ethical approval

The initial MCS protocol was approved by National Health Services (NHS) Research Ethics Committee (REC) of the South West. Further sweeps of data collection have also been approved by the NHS REC system. Data were pseudonymised to prevent participant identification and were subsequently made available at the UK Data Service platform, from where we accessed them (the data were anonymous to us). No formal ethical approval was required for this secondary analysis.

## Data collection

All data collection was carried out during home interviews conducted by trained study personnel using computer-assisted personal interviewing [37,43]. Except for the outcomes—which were cognitive tests led by the interviewer or self-completed by the child (as will be described in the following section)—, data for all variables was provided by the mother as the main interviewee (and the partner in the case of their education and social class) [43].

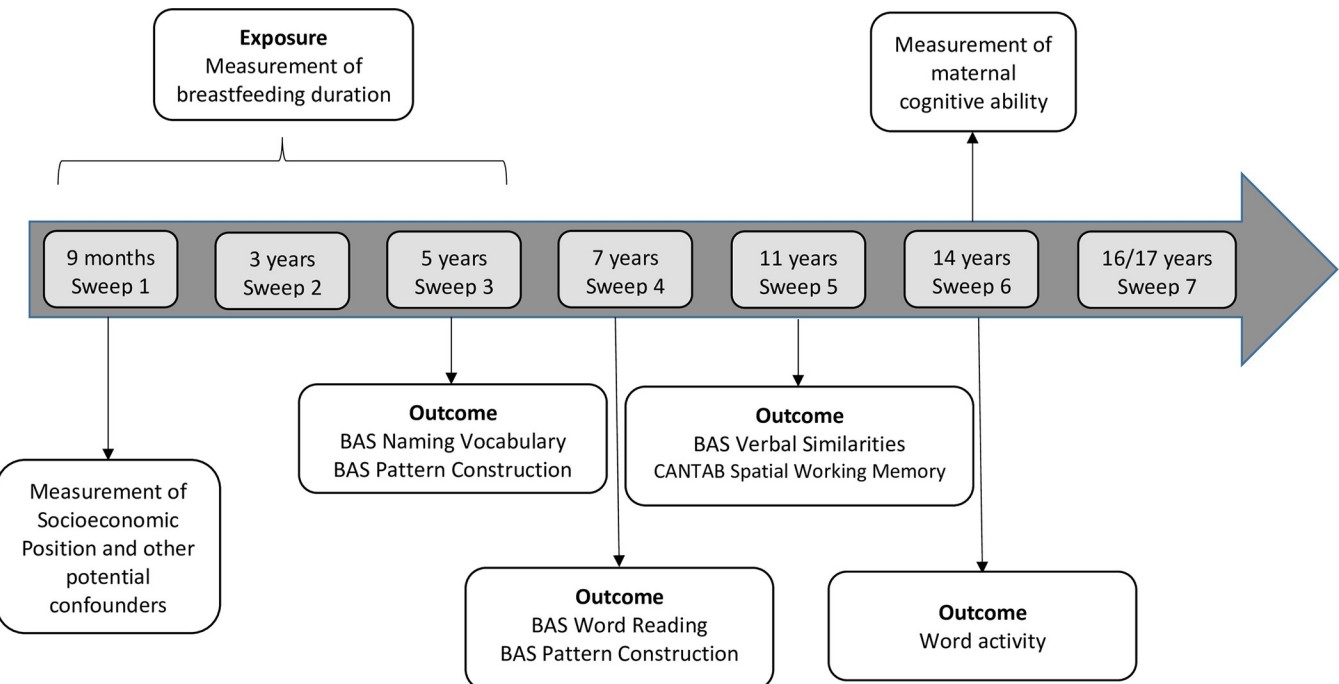

**Fig 1. Association between breastfeeding duration and cognitive development among children from the UK Millennium Cohort Study.**

## Outcomes

The outcomes of interest were variables evaluating two domains of cognitive development: verbal abilities and spatial awareness.

**Verbal cognitive development.** Verbal ability was evaluated at ages 5, 7, 11 and 14 years. It was measured using the British Ability Scales Second Edition (BAS II), at ages 5 (BAS Naming Vocabulary, evaluating expressive verbal ability and vocabulary knowledge), 7 (BAS Word Reading, evaluating knowledge of reading), and 11 (BAS Verbal Similarities, evaluating verbal reasoning and knowledge) [43]. These tasks were led by the interviewer. At age 14, a "Word activity" self-completion instrument (evaluating the understanding of the meaning of words) [44] was used (**Fig 1**). These instruments have appropriate construct validity, high test-retest reliability, and concurrent validity with other vocabulary tests [45,46].

**Spatial cognitive development.** Spatial ability was assessed at ages 5, 7 and 11 years [43]. The BAS Pattern Construction sub-test was administered at ages 5 and 7 and the Cambridge Neuropsychological Test Automated Battery (CANTAB] was administered at age 11 (**Fig 1**). The BAS test (interviewer-led) evaluates spatial problem solving skills [45] and the CANTAB is a self-completion tool that evaluates the ability to memorise spatial information [46], considering two aspects: i) 'Strategy' (how systematic was the child while executing the test) and ii) the number of 'Errors' incurred during the test.

**Standardisation.** To allow meaningful comparison of effect sizes across different time points, continuous scores were standardised (mean = 0; standard deviation = 1) by sex and age at measurement [three-month intervals], considering the distribution in all infants that were included in the analysis.

## Exposure

Breastfeeding duration was evaluated by maternal responses to the questions: "Did you ever try to breastfeed your baby?" and "How old was your baby when s/he last had breast milk?" at ages 9 months and 5 years of age (for those breastfeeding for longer than 9 months, approximately 10% of included participants). The latter captured the duration of any BF which was grouped as: Never BF; <2 months; ≥2 and <4 months; ≥4 and <6 months; ≥6 and <12 months; ≥12 months.

Duration of exclusive breastfeeding (EBF) was defined as the time in which the child was fed with breast milk only, and was generated using a combination of the duration of BF and the age in which formula, cow's milk, other types of milk and solid foods were first introduced. The final variable was classified as: Never BF; <2 months; ≥2 and <4 months, ≥4 months. Maternal report has shown to be a valid and reliable measure of breastfeeding duration [47,48], even up to age 6 years [48].

## Potential confounders

After a literature review, a Directed Acyclic Graph (DAG) [49] was created in order to systematically represent the potential confounding factors that would need to be considered in the analysis [1,2] (**Fig 2**).

i) SEP: The markers of SEP used were social class (highest occupation between both parents: Managerial/Professional, Intermediate and Semi-routine/Routine; according to the National Statistics Socioeconomic Classification NS-SEC [50]) and maternal education [according to the National Vocational Qualification NVQ standards: Higher—NVQ 4 and 5 (University degree), Medium—NVQ3 (GCE A-level, national qualifications typically attained at age 18), Lower— NVQ 1 and 2 (national qualifications typically attained at the end of compulsory Secondary Education, age 16), Other/overseas qualifications, and "No formal education". While social

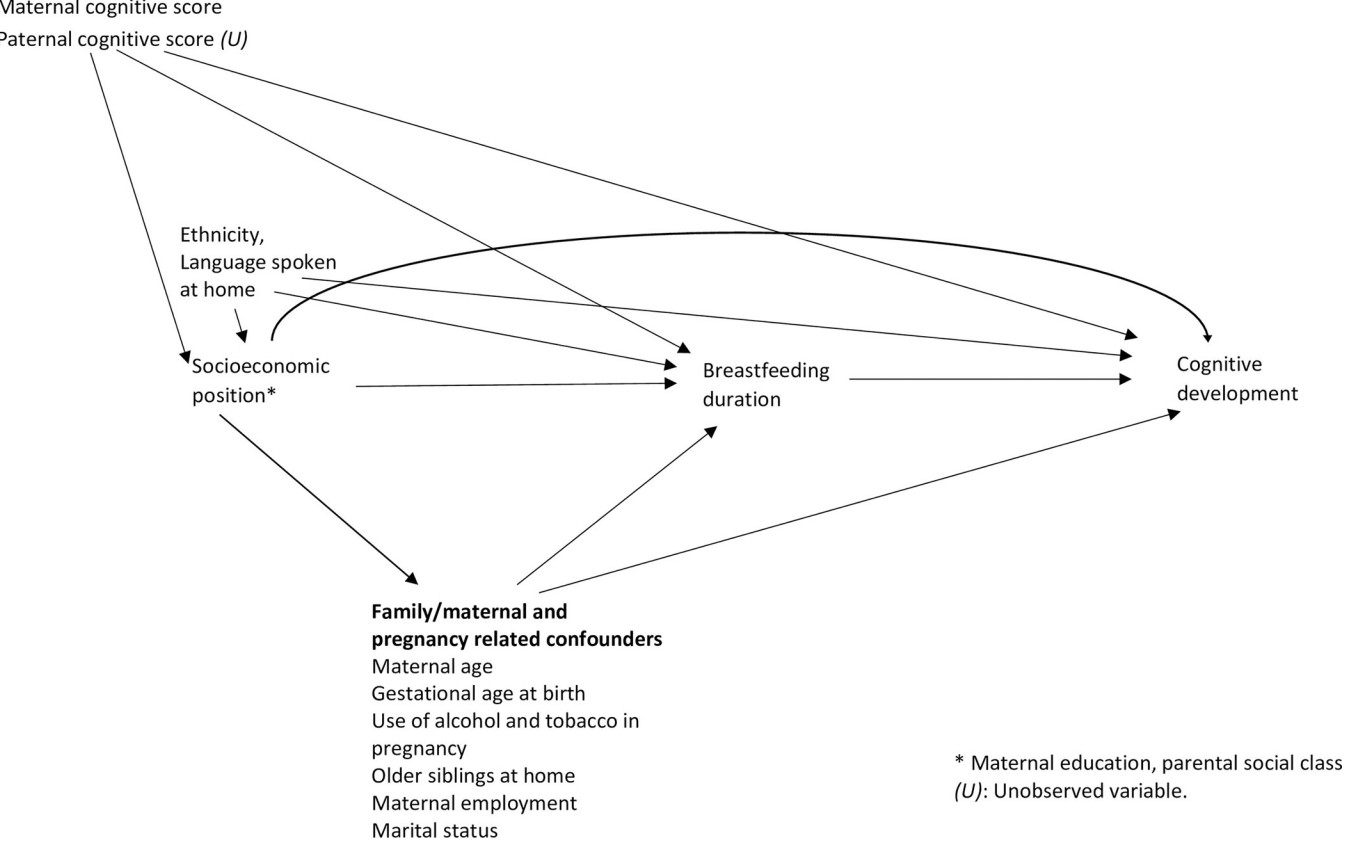

**Fig 2. Directed acyclic graph used for the evaluation of the association between breastfeeding duration and cognitive development.**

class and education are related, these two indicators capture different aspects of socioeconomic circumstances and adjusting for both reduces the risk of residual confounding [9].

ii) Other potential confounders (sociodemographic characteristics and variables related to pregnancy, childcare and health): Firstly, a basic set of confounders that included gestational age at birth (in weeks), maternal ethnicity (White vs Black, Asian and Minority Ethnic) and languages spoken in household (English only vs English and other language) was selected.

Other potential confounders included maternal partnership status (married, cohabiting, single mother), mother working outside the home (yes, no), birth by Caesarean section (yes, no), having older siblings in the household (yes, no), maternal age (in years), and alcohol and tobacco use during pregnancy (yes, no). All these variables were measured at 9 months of age.

iii) Maternal cognitive ability: MCS evaluated maternal verbal ability as a proxy for maternal intelligence, when the children were 14 years old. Mothers completed a "Word activity" questionnaire (similar to the Vocabulary test used for participants at age 14) [44]. This questionnaire was adapted from a standardised vocabulary test developed by the Applied Psychology Unit of the University of Edinburgh, and has been previously used in the 1970 British Birth Cohort [51,52]. These scores were standardised (mean = 0; standard deviation = 1).

## Statistical analysis

The association between BF duration and categorical variables was evaluated using the chi-square test and its association with continuous variables was evaluated using the F-test. Given that cognitive development was assessed across two different domains (spatial, verbal), these

were analysed separately. Crude and adjusted coefficients and 95% confidence intervals (95% CI) were calculated using unbalanced mixed-effects generalised linear models of the Gaussian family and identity link function, assuming an unstructured covariance matrix. The repeated measures of the outcome were clustered within each child. As the outcomes have been standardised, the coefficients are expressed in standard deviations (SD). The different durations of breastfeeding were compared to 'never breastfed' as the reference category [15]. Interactions between breastfeeding duration and the age at the time of cognitive assessment were fitted in all models and retained if they were statistically significant at the 5% level.

Adjustment for different sets of confounders was completed sequentially. Model 1 adjusted for a basic set of confounders selected *a priori* (maternal age, ethnicity and language spoken at home). In Model 2, SEP markers were added (regardless of their association with the outcome). Model 3 incorporated a set of potential confounders into the model (from the previously described pool of variables).

The process to select which potential confounders were included into the final models was guided by statistical criteria. Each potential confounder was added separately to Model 2 and remained if it showed an association with the outcome (p<0.10), after adjusting for other variables in the model. Adjustment for maternal cognitive scores was conducted as a final step (Model 4).

All analyses were conducted in Stata 15.0 for Windows [53] and accounted for the sampling design and attrition at each sweep using the Stata 'svy' modules [54] and study weights [55].

E-values were calculated to evaluate the potential for residual confounding to explain the observed associations in the final, fully adjusted models [56].

## Sensitivity analysis

All analyses were repeated using exclusive breastfeeding (EBF) duration (instead of any BF) as the exposure. Additionally, these associations might behave differently among children of White English-speaking mothers (a more homogeneous group that constitutes the majority of the UK population), so analyses restricted to these participants were conducted. Previous research has also explored similar research questions on the White British subpopulation of the MCS [16].

## Results

### Characteristics of the participants

A total of 7,855 and 7,582 participants were included in the analyses of verbal and spatial cognitive outcomes, respectively (**Fig 3**). Half (49.8%) of the children were female; 25.9% of children had mothers in the highest education group. The majority of the mothers were White (90.2%) and lived in households in which only English was spoken (93.3%) (**Table 1**).

### Breastfeeding duration

Approximately, 33.9% of participants were never breastfed and 23.0% were breastfed for six months or longer. Almost all covariates explored were strongly associated with breastfeeding duration (**Table 1**). In particular, the longer the BF duration, the higher the probability of having a more educated mother or a higher parental social class. Children who were breastfed for longer were more likely to have non-smoking, married, and older mothers. They were also less likely to be born to White and English-speaking mothers. Mean maternal cognitive scores were higher among women who breastfed their babies for longer.

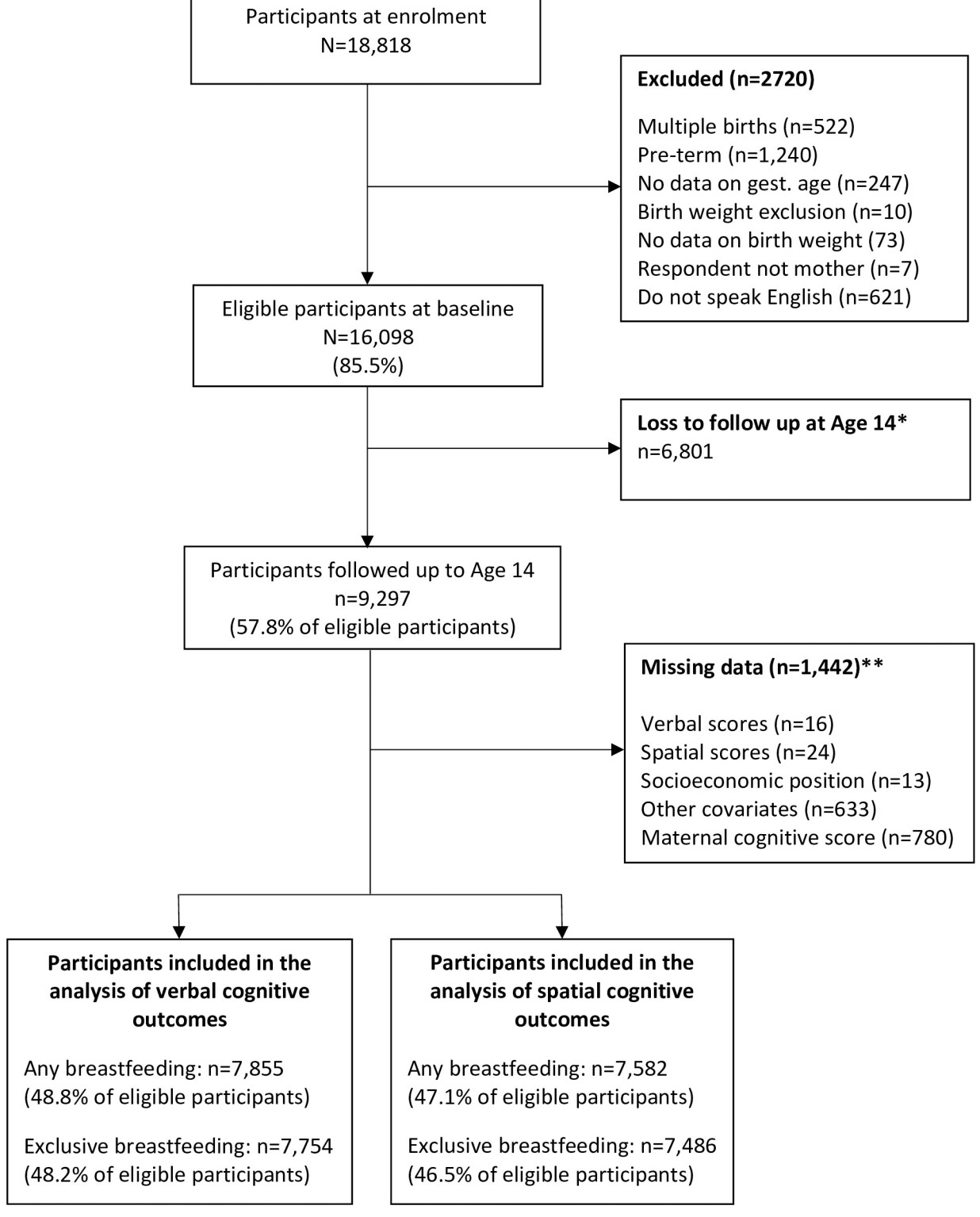

**Fig 3. Flowchart of study participants included in the analysis, UK Millennium Cohort Study.** *Those not present at age 14 were excluded because data for maternal cognitive scores were assessed during that evaluation. **Data missing for the analysis of verbal scores.

**Table 1. Characteristics of the participants according to BF duration, UK Millennium Cohort Study (n = 7,855).**

| Characteristics | Descriptive Whole sample (n = 7,855) | | Any BF duration | | | | | | | | | | | | |
|---|---|---|---|---|---|---|---|---|---|---|---|---|---|---|---|
| | | | Never BF (n = 2,179) | | <2 months (n = 2,056) | | ≥2 to <4 mo (n = 819) | | ≥4 to <6 mo (n = 784) | | ≥6 to <12 mo (n = 1,175) | | ≥12 months (n = 842) | | |
| | n | (%) | n | (%) | n | (%) | n | (%) | n | (%) | n | (%) | n | (%) | P * |
| **Pregnancy and child-related** | | | | | | | | | | | | | | | |
| Gestational age at birth | 39.8 | (1.3) | 39.8 | (1.2) | 39.8 | (1.4) | 39.8 | (1.3) | 39.9 | (1.3) | 39.9 | (1.4) | 39.9 | (1.3) | 0.163 |
| Female | 4004 | (49.8) | 1158 | (51.9) | 989 | (46.9) | 421 | (51.2) | 383 | (45.3) | 601 | (48.5) | 452 | (54.0) | <0.001 |
| Age at cognitive test (in months, at age 5)$^\S$ | 63.3 | (3.1) | 63.3 | (2.8) | 63.3 | (3.2) | 63.3 | (3.1) | 63.3 | (3.1) | 63.3 | (3.2) | 63.2 | (3.3) | 0.845 |
| No older siblings | 3873 | (48.7) | 1175 | (51.9) | 842 | (40.5) | 359 | (44.2) | 372 | (45.8) | 646 | (54.7) | 479 | (56.6) | <0.001 |
| Smoked during pregnancy | | | | | | | | | | | | | | | |
| Never | 5439 | (64.3) | 1201 | (50.7) | 1341 | (60.6) | 573 | (65.2) | 613 | (75.9) | 973 | (81.3) | 738 | (86.0) | <0.001 |
| Gave up | 943 | (13.2) | 266 | (13.1) | 301 | (16.1) | 118 | (16.6) | 93 | (12.1) | 99 | (9.3) | 66 | (8.9) | |
| Kept smoking | 1473 | (22.5) | 712 | (36.2) | 414 | (23.3) | 128 | (18.2) | 78 | (12.0) | 103 | (9.4) | 38 | (5.1) | |
| Mod/heavy alcohol in pregnancy | 561 | (7.3) | 167 | (7.5) | 131 | (6.9) | 55 | (6.3) | 57 | (7.1) | 93 | (8.5) | 58 | (7.5) | 0.690 |
| **Sociodemographic** | | | | | | | | | | | | | | | |
| Maternal age (years)$^\S$ | 29.9 | (6.1) | 27.7 | (5.6) | 29.5 | (6.1) | 30.4 | (5.9) | 31.7 | (5.5) | 32.3 | (5.5) | 32.8 | (5.7) | <0.001 |
| Maternal education | | | | | | | | | | | | | | | |
| Higher | 2572 | (25.9) | 279 | (8.6) | 602 | (23.9) | 280 | (28.9) | 379 | (41.6) | 603 | (46.4) | 429 | (45.5) | <0.001 |
| Medium | 864 | (9.5) | 162 | (5.6) | 249 | (9.9) | 109 | (12.4) | 98 | (11.7) | 144 | (12.1) | 102 | (13.8) | |
| Lower | 3395 | (47.5) | 1236 | (5.7) | 969 | (51.4) | 359 | (49.8) | 249 | (35.8) | 349 | (34.3) | 233 | (30.0) | |
| Other | 149 | (1.9) | 37 | (1.4) | 24 | (1.4) | 20 | (2.5) | 17 | (2.6) | 27 | (2.1) | 24 | (3.4) | |
| None | 875 | (15.2) | 465 | (27.2) | 212 | (13.4) | 59 | (6.4) | 41 | (6.4) | 52 | (5.1) | 54 | (7.3) | |
| Highest social class | | | | | | | | | | | | | | | |
| Managerial/professional | 3945 | (44.2) | 534 | (25.8) | 807 | (44.8) | 377 | (53.4) | 456 | (63.7) | 711 | (68.7) | 489 | (65.3) | <0.001 |
| Intermediate | 2151 | (28.0) | 572 | (31.5) | 516 | (32.8) | 184 | (27.2) | 149 | (23.6) | 187 | (21.4) | 149 | (22.8) | |
| Semi-routine/routine | 1478 | (22.8) | 520 | (35.2) | 291 | (18.9) | 99 | (16.8) | 53 | (9.6) | 69 | (8.0) | 58 | (9.3) | |
| Not applicable | 281 | (5.0) | 91 | (7.5) | 47 | (3.5) | 15 | (2.6) | 12 | (3.1) | 17 | (1.9) | 18 | (2.6) | |
| Mother working at age 9 m | 4247 | (50.0) | 970 | (39.9) | 1162 | (53.0) | 492 | (59.7) | 507 | (61.6) | 708 | (57.4) | 408 | (46.5) | <0.001 |
| Maternal partnership status | | | | | | | | | | | | | | | |
| Married | 5064 | (57.3) | 1064 | (40.3) | 1296 | (58.5) | 555 | (63.7) | 570 | (66.7) | 919 | (74.5) | 660 | (74.3) | <0.001 |
| Cohabitation | 1846 | (27.3) | 640 | (34.0) | 505 | (27.5) | 192 | (25.5) | 164 | (24.3) | 207 | (20.1) | 138 | (17.7) | |
| Single mother | 945 | (15.4) | 475 | (25.7) | 255 | (14.0) | 72 | (10.8) | 50 | (8.9) | 49 | (5.5) | 44 | (8.0) | |
| Maternal ethnicity: White | 7007 | (90.2) | 2068 | (95.5) | 1851 | (91.0) | 709 | (87.0) | 659 | (82.9) | 1031 | (88.8) | 689 | (82.3) | <0.001 |
| Language spoken at home | | | | | | | | | | | | | | | |
| English only | 7137 | (93.3) | 2061 | (96.2) | 1887 | (94.2) | 738 | (92.0) | 678 | (90.1) | 1061 | (92.6) | 712 | (86.6) | <0.001 |
| English + other language | 718 | (6.7) | 118 | (3.8) | 169 | (5.8) | 81 | (8.0) | 106 | (9.9) | 114 | (7.4) | 130 | (13.4) | |
| **Maternal cognitive score**$^\S$ | -0.10 | (1.04) | -0.55 | (0.79) | -0.15 | (1.01) | -0.02 | (0.96) | 0.27 | (1.07) | 0.37 | (1.09) | 0.41 | (1.20) | <0.001 |
| **Verbal cognitive scores** | | | | | | | | | | | | | | | |
| BAS Vocabulary, Age 5$^\S$ (n = 7805) | -0.05 | (1.04) | -0.26 | (0.90) | -0.03 | (1.08) | 0.04 | (1.03) | 0.05 | (1.11) | 0.20 | (1.11) | 0.13 | (1.08) | <0.001 |
| BAS Word Read, Age 7$^\S$ (n = 7735) | -0.03 | (1.03) | -0.28 | (0.94) | -0.05 | (1.04) | 0.06 | (1.05) | 0.23 | (1.01) | 0.21 | (1.04) | 0.26 | (1.01) | <0.001 |
| BAS Verbal Similarities, Age 11$^\S$ (n = 7510) | -0.08 | (1.04) | -0.28 | (0.96) | -0.11 | (1.08) | -0.07 | (1.12) | 0.14 | (0.98) | 0.16 | (0.99) | 0.18 | (0.96) | <0.001 |
| Word Activity, Age 14$^\S$ (n = 7376) | -0.05 | (1.03) | -0.29 | (0.84) | -0.11 | (1.00) | -0.03 | (1.01) | 0.16 | (1.09) | 0.25 | (1.16) | 0.33 | (1.25) | <0.001 |
| **Spatial cognitive scores**$^\S$ | | | | | | | | | | | | | | | |

*(Continued)*

**Table 1.** (Continued)

| | | | | | | | | | | | | | | |
|---|---|---|---|---|---|---|---|---|---|---|---|---|---|---|
| BAS Pattern Construction. Age 5 (n = 7,511)§ | -0.02 | (1.03) | -0.16 | (0.97) | -0.03 | (1.09) | 0.02 | (0.94) | 0.15 | (1.00) | 0.14 | (1.10) | 0.06 | (1.02) | <0.001 |
| BAS Pattern Construction. Age 7 (n = 7,491)§ | -0.05 | (1.04) | -0.24 | (0.93) | -0.06 | (1.11) | 0.00 | (1.04) | 0.12 | (1.09) | 0.16 | (1.07) | 0.13 | (1.00) | <0.001 |
| CANTAB Spat.—Strategy, Age 11 (n = 7,296)§ | 0.00 | (1.04) | -0.15 | (0.84) | -0.05 | (1.02) | 0.02 | (1.03) | 0.19 | (1.24) | 0.16 | (1.19) | 0.17 | (1.24) | <0.001 |
| CANTAB Spat.—Errors, Age 11 (n = 7,296)§ | 0.00 | (1.04) | -0.18 | (0.93) | -0.04 | (1.07) | 0.04 | (1.06) | 0.25 | (1.07) | 0.17 | (1.10) | 0.17 | (1.13) | <0.001 |

§mean (SD) for numerical variables.

*chi$^2$ / F test.

These estimates consider the complex sampling design.

The frequencies are unweighted counts, and the percentages are weighted using design and non-response weights.

## Verbal scores

Both maternal education and parental social class showed marked graded associations with verbal scores across the four sweeps. There were positive correlations between all verbal scores and the maternal cognitive score (S1 Table).

**Association between breastfeeding duration (any BF) and verbal scores.** In the multivariable linear models, there was an interaction between breastfeeding duration and the age at which the verbal cognitive scores were evaluated (p<0.001). Therefore, the models retained the interaction term and coefficients for the outcome at each age are presented.

The crude analysis at age 5 indicated that longer BF durations were associated with higher average BAS Vocabulary scores (Fig 4). Children who were breastfed for ≥12 months had an average score 0.39 SD (95%CI: 0.30 to 0.47) higher than those never breastfed. After controlling for maternal age, ethnicity and the language spoken in the household (Model 1), the regression coefficients showed a slight increase. Adjustment for SEP markers (Model 2) reduced the effect sizes by approximately half (0.15 SD; 95%CI: 0.08 to 0.23, ≥12 months vs never breastfed). Further adjustment for the remaining potential confounders (Model 3) did not markedly change the coefficients. Further adjustment for maternal cognitive scores (Model 4) attenuated almost all coefficients to values that were not significantly different from zero (0.03 SD; 95%CI: -0.04 to 0.10, ≥12 months vs never breastfed).

At age 7, adjustment for SEP and the rest of the confounders had a similar effect as at age 5. However, controlling for maternal cognitive scores did not fully explain the association of interest. In the fully-adjusted model, longer breastfeeding durations were associated with higher cognitive verbal scores in children (0.19 SD; 95%CI: 0.11 to 0.27, ≥12 months vs never breastfed).

At ages 11 and 14, children breastfed for ≥6 months scored higher in verbal cognitive scores in comparison to children never breastfed, even after controlling for all confounders, including maternal cognitive scores. However, at age 11, the coefficients were smaller (0.08 SD; 95%CI: 0.00 to 0.16, ≥12 months vs never breastfed) than at age 14 (0.26 SD; 95%CI: 0.18 to 0.34, ≥12 months vs never breastfed) (Fig 4).

## Spatial scores

Both maternal education and parental social class showed graded positive associations with spatial scores in the three sweeps. All the spatial tests administered were positively correlated with maternal cognitive scores (S2 Table).

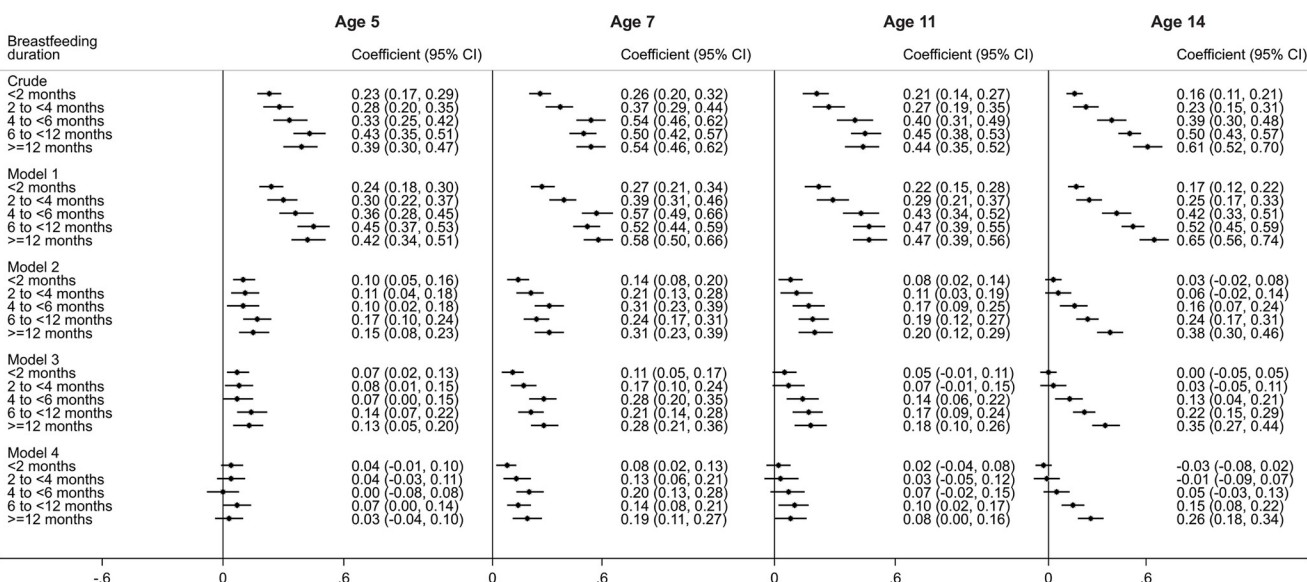

**Fig 4. Association between breastfeeding duration (any breastfeeding) and standardised cognitive verbal scores (mean: 0; SD: 1)between ages 5 and 14, UK Millennium Cohort Study (n = 7,855).** All categories of BF duration are compared to "Never breastfed" as the reference category. Model 1: Adjusted for gestational age at birth, maternal ethnicity and languages spoken in household. Model 2: Adjusted for Model 1 + Socioeconomic position (maternal education and highest social class in household). Model 3: Adjusted for Model 2 + other confounding factors (older siblings in household, maternal age, mother working outside the home, partnership status, and maternal smoking during pregnancy). Model 4: Adjusted for Model 4 + Maternal cognitive score.

**Association between breastfeeding duration (any BF) and spatial scores.** The interaction between the exposure and the age at which spatial scores were assessed (p<0.001) was retained and age-specific coefficients are presented. The crude analysis shows that longer BF durations were associated with higher average BAS Pattern Construction scores at age 5, although there was less of a gradient compared with verbal scores (**Fig 5**). Children who were breastfed for 12 months or more had an average score 0.21 SD (95%CI: 0.13 to 0.29) higher than those never breastfed. After controlling for maternal age, ethnicity and the language spoken in the household, the regression coefficients again showed a slight increase. Adjustment for SEP markers considerably reduced the effect sizes and further adjustment for the remaining set of potential confounders only explained a small fraction of the associations. Adjustment for maternal cognitive scores considerably attenuated the estimates towards the null value. After full adjustment, there was not a clear gradient between BF durations and spatial scores. Those children who BF for 4 to <6 months had the highest average score, 0.11 SD (95%CI: 0.03 to 0.19) higher than those who were never breastfed. Meanwhile, the average score of those who were breastfed for ≥12 months was not different from the score of those never breastfed (-0.01 SD; 95%CI: -0.09 to 0.07).

Results for ages 7 and 11 showed that longer BF durations were associated with higher average spatial cognitive scores, even after full adjustment (**Fig 5**). Those breastfeeding for 4 to <6 also had the highest mean scores in comparison to those never breastfed. At age 11, the association was only present in the "Errors" Dimension of the CANTAB (as opposed to the "Strategy" Dimension, in which the coefficients were not statistically different from the null value).

## Relative effect of key confounding factors and potential for residual confounding

To put the coefficients for the association between breastfeeding duration and the child's cognitive scores into context, the coefficients for other variables that are associated with cognitive

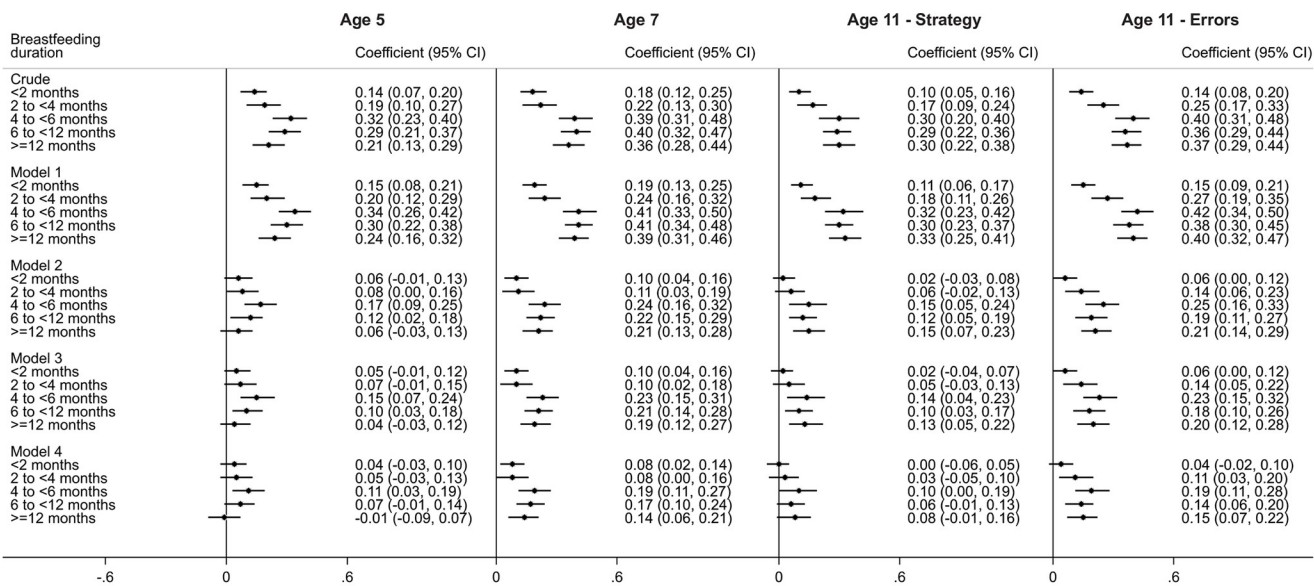

**Fig 5. Association between breastfeeding duration (any breastfeeding) and standardised cognitive spatial scores (mean: 0; SD: 1) between ages 5 and 11, UK Millennium Cohort Study (n = 7,582).** All categories of BF duration are compared to "Never breastfed" as the reference category. Model 1: Adjusted for gestational age at birth, maternal ethnicity and languages spoken in household. Model 2: Adjusted for Model 1 + Socioeconomic position (maternal education and highest social class in household). Model 3: Adjusted for Model 2 + other confounding factors (older siblings in household, mother working outside the home, partnership status, maternal alcohol use during pregnancy and smoking during pregnancy). Model 4: Adjusted for Model 3 + Maternal cognitive score.

scores in children, such as maternal education (high vs low) or parental social class (high vs low) were approximately 0.20 in the fully adjusted models of our analysis. Meanwhile, a one-SD increase in the standardised maternal cognitive score was associated with a 0.21 SD increase in the child's cognitive verbal score (**Fig 6**).

To account for the observed associations found in this study (for example, a coefficient of 0.26 for breastfeeding ≥12 months for verbal scores at age 14, or 0.10 for 4 to <6 months for spatial scores at age 11), any given unmeasured confounder should have a coefficient of at least 0.68 or 0.39 (which corresponds to E-values of 1.85 or 1.43 on the risk ratio scale), respectively, with both exposure and outcome (**S3 Table**).

## Sensitivity analysis

The analyses using EBF duration showed a similar pattern to those of any BF duration, as shown in **S1 Fig** (verbal scores) **and S2 Fig** (spatial scores).

The analyses carried out on the restricted sample of children of white, English-speaking mothers yielded similar conclusions (**S4** to **S7 Tables**).

## Discussion

### Summary of key findings

This study assessed how much of the association between breastfeeding duration and cognitive development is due to confounding by SEP and maternal cognitive scores, based on the data of a nationally representative UK cohort study. The unadjusted associations showed that longer BF durations were associated with higher verbal and spatial cognitive scores up to ages 14 and 11, respectively. Adjustment for SEP explained approximately half of the initially observed associations. Further adjustment for maternal cognitive measures failed to completely remove

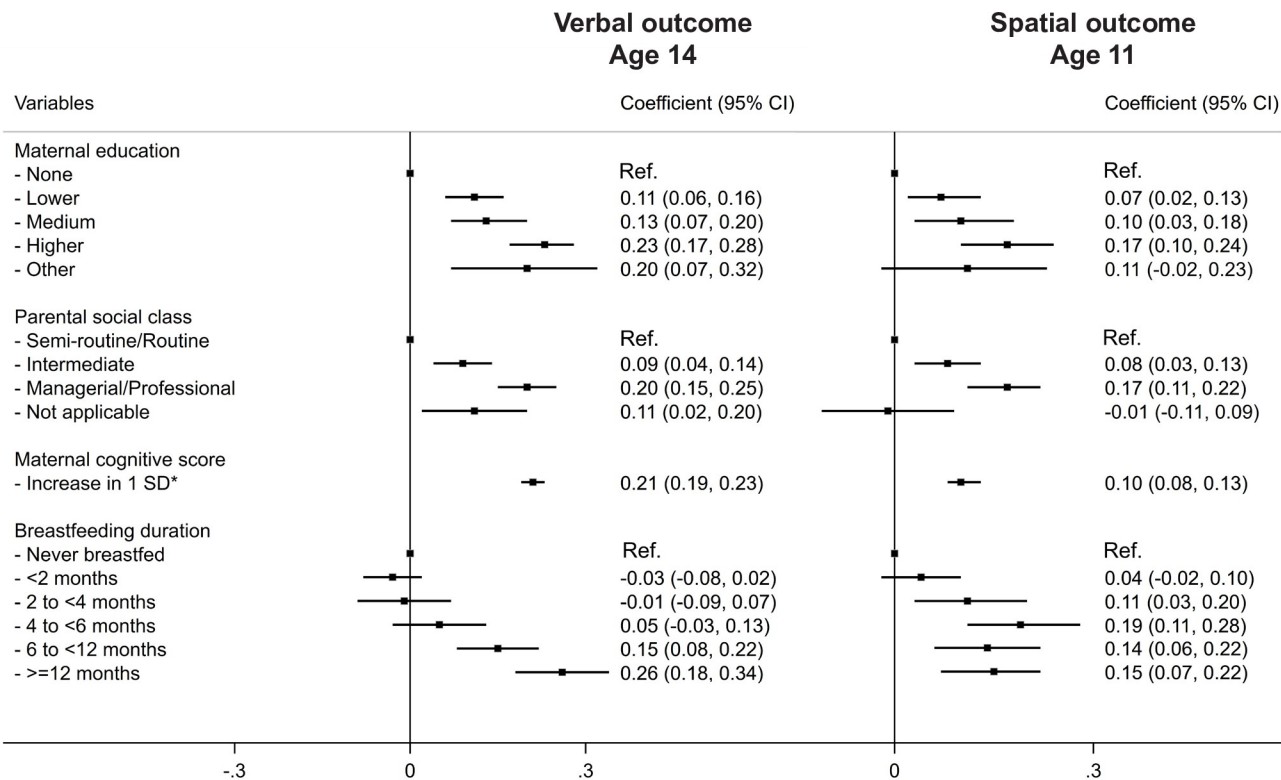

**Fig 6. Comparison of the coefficients for breastfeeding duration, markers of socioeconomic position and maternal cognitive scores on verbal and spatial cognitive outcomes* at ages 14 and 11, respectively.** The models adjusted for all potential confounders of the association between breastfeeding duration and cognitive scores, and include an interaction between breastfeeding duration and age of outcome measurement. *Standardised maternal cognitive scores (Mean: 0; SD: 1).

the remaining associations at ages 7, 11 and 14. The fully-adjusted coefficients where there is evidence of an effect of breastfeeding on verbal cognitive scores varied between 0.08 (age 7; <2 months vs never breastfed) to 0.26 SD (age 14; ≥12 months vs never breastfed). For spatial scores, the coefficients varied between 0.08 (age 7; <2 months vs never breastfed) to 0.19 SD (ages 7 and 11; 4 to <6 months vs never breastfed). This suggests that while the association in this population is not completely due to confounding, the effect of breastfeeding on cognitive development is modest in this population.

## The confounding effect of SEP and maternal cognitive scores

There are biologically plausible mechanisms through which breastfeeding could improve cognitive outcomes, such as the provision of myelination-inducing and neurodevelopment-enhancing long chain polyunsaturated fatty acids (PUFAs) and micronutrients (such as iron, folate, zinc, choline, among others). Human milk also contains microRNAs (miRNAs) that may be involved in epigenetic processes that promote the development of the brain and its functions [14,19,29,57]. Other potential mechanisms include the reduction of the risk of school absenteeism through protection from infectious diseases and the maternal attachment secondary to contact during breastfeeding [2,14,29]. However, it has also been argued that the observed associations are due to confounding [1–3]. SEP has been extensively described as one of the main confounders of this association [1–4]. Women from a higher SEP tend to have higher "health literacy", be more receptive to health education, have better maternity benefits/

working conditions, and stronger social networks [8,9], all of which could influence their decisions/ability to breastfeed. In this study, adjustment for SEP explained approximately half of the observed effect. This is consistent with systematic reviews on the topic which have found that adjustment for SEP, on average, reduces the effect sizes by a similar magnitude [1–3]. Despite this reduction, adjustment for SEP did not completely remove the observed associations in our study. While most studies that only control for SEP (and not maternal intelligence) also show positive associations with cognitive development in childhood [13–19], some others, such as two large cohort studies among Irish [20] and British [21] children found no association after adjusting for SEP and other sociodemographic confounders.

This study found that adjustment for maternal cognitive measures explained a considerable proportion of the remaining associations, but did not remove them at ages 7, 11 and 14. As with SEP, higher maternal intelligence could favour better uptake of health information and consequently, increase the probability/duration of BF [1–3,23]. Prior evidence on its confounding effect is more heterogeneous than SEP. Several studies have found that adjustment for maternal measures of intelligence (in addition to SEP) completely removes the initially observed associations [20,21,24]. On the other hand, other studies found persistently significant positive associations after adjustment for this variable [26–31], albeit with very small effect sizes [28,29]. A 2015 meta-analysis pooled the estimates of the studies that controlled for maternal intelligence and found a positive association [1], however it has been criticised for overestimating the effect size due to sub-optimally addressed publication bias [58]. It is important to note that most of the previous reports have traditionally dichotomised breastfeeding duration as yes/no or with a temporal cut-off, which may hide important information. Conversely, our study uses several categories of duration, which helps to explore this relationship in a more nuanced way.

### The association between breastfeeding duration and cognitive scores

Longer breastfeeding durations were associated with mean cognitive scores that were 0.08 and 0.26 SD higher than the mean cognitive score of those never breastfed. The meta-analysis by Horta et al. reported that, among those studies that controlled for maternal IQ, the pooled effect size was 2.62 IQ points (95% CI: 1.25, 3.98) [1]. Given that the IQ scale is expressed as a mean of 100 with an SD of 15, the coefficients in our study are similar to these combined estimates. While coefficients of 0.08 to 0.26 correspond to modest increases in the cognitive scores, the strength of this association should not be overlooked, as it may be comparable to the strength of the association for other recognised predictors of cognitive development, such as SEP and maternal cognitive ability in these models.

We also considered the possibility that the remaining associations were explained by residual confounding produced by unmeasured confounders, such as paternal measures of cognitive ability or broader measures of maternal cognitive ability. This was assessed through the calculation of the E-values [56]. In order to explain the aforementioned associations, any unmeasured confounder should be associated with both BF duration and cognitive scores with coefficients of at least 0.39 (to fully explain a coefficient of 0.10) or 0.68 (to fully explain a coefficient of 0.26). Therefore, while there is room for the associations to be further explained, it is unlikely that all the observed associations could be explained in full by additional adjustment.

Some findings deserve further investigation. The association at age 14 seems to be stronger than at other ages. The outcome was measured with a different instrument at age 14, which may contribute to the observed differences. However, these results seem to be in line with those of Kanazawa, who showed that the effect of BF on intelligence increased as children got older in the 1958 British Birth cohort [31]. On the other hand, the follow-up evaluation at age 16 among the sample from the PROBIT experimental study found that the cognitive benefits

initially seen at younger ages largely disappeared, except for a modest effect in verbal scores [59]. Additionally, the association with spatial outcomes did not follow a gradient in our study. Sajjad et al. found a similar pattern in a Dutch cohort that evaluated the association between breastfeeding duration and non-verbal intelligence [60]. However, these findings could be due to chance or (less likely) to residual confounding and should be revisited in future studies.

It is important to mention that, like several previous studies [1–3], we compare those who breastfed for a given period of time, versus those who were not breastfed (or were breastfed for a shorter duration in some studies). This, however, may not be comparable across studies and could partly explain the heterogeneity in the findings. Not breastfeeding/breastfeeding for shorter periods of time would entail using either formula, cow's milk or other types of liquids and foods; and these would largely depend on the setting. Moreover, within formula-users, the composition of the formula will probably vary according to the geographical and temporal context [61]. Therefore, studies in different countries and time frames will likely differ in terms of what it means to "not be breastfed", which should be considered when interpreting the results.

## Limitations and strengths

There are some limitations that should be considered when interpreting the findings of the present study. Maternal cognitive tests in the MCS evaluate their understanding of the meaning of several words, which could be affected by further education and might not necessarily reflect broader intelligence. However, similar measures seem to be correlated with verbal ability and have been used as a proxy for intelligence in previous studies [1,2]. Previous studies have produced conflicting results irrespective of whether they have adjusted for verbal or global measures of maternal intelligence, [23–31]. Also, our results indicate that this variable explains the association more than maternal education alone, which suggests that this variable is capturing more than just educational attainment. Ideally, future studies should assess maternal intelligence at baseline using validated multi-dimensional intelligence tests.

The proportion of participants who were excluded from the analysis due to missing data for covariates (item non-response) was approximately 15%. The variable that accounted for the highest proportion of missing data was the maternal cognitive measure (approximately 8%). However, non-participation (unit non-response) resulted in the exclusion of approximately half of the original cohort. This attrition was corrected by using non-response survey weights (adjusting for socio-demographic and clinical factors associated with non-response at baseline and subsequent surveys), thus minimising the effect of selection bias due to loss to follow up [55], which is a recommended approach when dealing with this frequent scenario in longitudinal studies [62].

This study also has important strengths. Using in a nationally-representative longitudinal study with a large sample size, we evaluated cognitive outcomes up to age 14 among the same participants thus allowing a comparison of the effects through childhood and early adolescence. Additionally, our study also evaluated two different aspects of cognition, and the confounding effect was apparent in both. Moreover, the findings using the duration of 'any BF' as the exposure of interest were confirmed in the analysis of exclusive breastfeeding (EBF). Lastly, adjustment for SEP included both social class and education, leaving little room for residual confounding by social circumstances [9].

## Conclusions

In conclusion, the positive associations between breastfeeding duration and cognitive development up to age 14 among children from the MCS were not explained in full after adjusting for SEP and maternal cognitive scores. While the size of the fully adjusted coefficients for breastfeeding duration was modest, they were comparable to the coefficients for SEP markers and

maternal cognitive scores. This suggests that the role of breastfeeding on the child's cognitive scores should not be underestimated. While a small increase in cognitive outcomes may not be clinically meaningful at the individual level, it has the potential to be influential at the population level. All future studies should ensure proper control of both socioeconomic factors and maternal intelligence. Breastfeeding should continue to be encouraged, as any improvements in child's cognitive are only one aspect of the benefits it provides.

## Supporting information

**S1 Fig. Association between exclusive breastfeeding duration*** **and standardised cognitive verbal scores between ages 5 and 14, UK Millennium Cohort Study (n = 7,754).**
(PDF)

**S2 Fig. Association between exclusive breastfeeding duration*** **and standardised cognitive spatial scores between ages 5 and 11, UK Millennium Cohort Study (n = 7,068).**
(PDF)

**S1 Table. Verbal cognitive scores according to the characteristics of the study subjects, UK Millennium Cohort Study (n = 7,855).**
(DOCX)

**S2 Table. Spatial cognitive scores according to the characteristics of the study subjects, UK Millennium Cohort Study (n = 7,582).**
(DOCX)

**S3 Table. Calculation of E-values: Association between breastfeeding duration (any breastfeeding) and cognitive development, UK Millennium Cohort Study.**
(DOCX)

**S4 Table. Association between breastfeeding duration (any breastfeeding) and standardised cognitive verbal scores (mean: 0; SD: 1) between ages 5 and 14 among children of white English-speaking mothers, UK Millennium Cohort Study (n = 6,834).**
(DOCX)

**S5 Table. Association between breastfeeding duration (any breastfeeding) and standardised cognitive spatial scores (mean: 0; SD: 1) between ages 5 and 11 among children of white English-speaking mothers, UK Millennium Cohort Study (n = 6,608).**
(DOCX)

**S6 Table. Association between exclusive breastfeeding duration*** **and standardised cognitive verbal scores (mean: 0; SD: 1) between ages 5 and 14 among children of white English-speaking mothers, UK Millennium Cohort Study (n = 6,752).**
(DOCX)

**S7 Table. Association between exclusive breastfeeding duration*** **and standardised cognitive spatial scores (mean: 0; SD: 1) between ages 5 and 11 among children of white English-speaking mothers, UK Millennium Cohort Study (n = 6,530).**
(DOCX)

## Acknowledgments

The authors would like to thank the children and families who participated in the Millennium Cohort Study and the UK Data Service for providing the datasets. We would also like to thank Sarah Chamberlain for her help in producing the figures.

## Presentation at meeting

This work has been presented at the Annual Scientific Meeting of the Society for Social Medicine & Population Health (Liverpool, 2021). Abstract: https://jech.bmj.com/content/75/Suppl_1/A44.3

## Author Contributions

**Conceptualization:** Reneé Pereyra-Elías, Maria A. Quigley, Claire Carson.

**Formal analysis:** Reneé Pereyra-Elías, Maria A. Quigley, Claire Carson.

**Methodology:** Reneé Pereyra-Elías, Maria A. Quigley, Claire Carson.

**Supervision:** Maria A. Quigley, Claire Carson.

**Writing – original draft:** Reneé Pereyra-Elías.

**Writing – review & editing:** Maria A. Quigley, Claire Carson.

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
