## [Decision Letter · Decision Letter 0]

15 Mar 2022

PONE-D-22-03469To what extent does confounding explain the association between breastfeeding duration and cognitive development up to age 14?  Findings from the UK Millennium Cohort StudyPLOS ONE

Dear Dr. Pereyra-Elías,

Thank you for submitting your manuscript to PLOS ONE. After careful consideration, we feel that it has merit but does not fully meet PLOS ONE’s publication criteria as it currently stands. Therefore, we invite you to submit a revised version of the manuscript that addresses the points raised during the review process.

The authors present a longitudinal analysis of a large cohort of children, examining the influence of confounding on the relationship between breast-feeding variables and cognitive outcomes. The manuscript is clear and well-written and the questions addressed represent an important contribution to the field.

As noted in the reviews, please could you provide further clarity on the methodologies of data collection as well as a discussion of the potential bias inherent in the tests of maternal intelligence used; and the difficulties of representing the breast-feeding variable. Other points of clarity have been raised in the reviewers’ comments.

We look forward to receiving your revised manuscript.

Kind regards,

Emma K. Kalk

Academic Editor

PLOS ONE

Journal Requirements:

2. In your ethics statement in the manuscript and in the online submission form, please provide additional information about the data used in your retrospective study. Specifically, please ensure that you have discussed whether all data were fully anonymized before you accessed them.

Reviewers' comments:

Reviewer's Responses to Questions

**Comments to the Author**

1. Is the manuscript technically sound, and do the data support the conclusions?

Reviewer #1: Yes

Reviewer #2: Yes

Reviewer #3: Yes

2. Has the statistical analysis been performed appropriately and rigorously? 

Reviewer #1: Yes

Reviewer #2: Yes

Reviewer #3: Yes

3. Have the authors made all data underlying the findings in their manuscript fully available?

Reviewer #1: Yes

Reviewer #2: Yes

Reviewer #3: Yes

4. Is the manuscript presented in an intelligible fashion and written in standard English?

Reviewer #1: Yes

Reviewer #2: Yes

Reviewer #3: Yes

5. Review Comments to the Author

Reviewer #1: This study examined the relationship between breastfeeding duration and cognitive development until children were 14 years old using a longitudinal data of a UK cohort study. Verbal and spatial abilities of the children were repeatedly assessed at 5, 7, 11 and 14 years old. The results indicate that the associations were still positive at 14 years old even after controlling for full confounding factors measured.

Although the manuscript includes a number of valuable data on the beneficial effects of breastfeeding, there were several points that should be clarified.

[Introduction]

(1) L59: (spell error?) ‘ould’ may be ‘could’ or ‘would’.

[Methods]

-Exposure

(2) L111: How the study group collected the data of breastmilk, formula, solid foods, etc. The authors stated ‘Breastfeeding duration was evaluated by maternal responses to the questions: “Did you …”.’. Did they asked the questions when they visited a research centre for child’s assessment or received questionnaires afterwards by mail. The methodologies of the investigation could affect mother’s responses.

-Potential confounders

(3) In DAG (Fig.2), there were a space between ‘Maternal cognitive score’ and ‘Paternal cognitive score’. I first missed the above. The authors may want to remove the space.

–Statistical analysis

(4) L156: They used mixed-effects generalised linear models for explaining the repeated measures of cognitive assessments within each child. However, there seem to be no results of overall verbal or spatial ability. Please describe it more clearly.

(5) L163: “Models 1 and 2 adjusted for a basic set of confounders (maternal age, ethnicity and language spoken at home) and SEP markers, respectively.” This could mean that model 1 adjusted for a basic set of confounders but not for SEP while model 2 adjusted for SEP but not for the basic set. However, actually, in the legends of Fig.4 and 5, “Model 2 adjusted for Model 1 + SEP”. Please describe them consistently.

[Results]

(6) In Table 1, ‘§’ may be missed at ‘Gestational age at birth’; ‘Maternal age (years)’; the four items of ‘Spatial cognitive scores’, in the ‘Characteristics’ column.

(7) L231-234: The authers may want to write like this, “… and the language spoken in the household (model 1), … Further adjustment for maternal cognitive scores (model 4) attenuated …”. The descriptions are a bit hart to follow as it was.

(8) L322: The authors may write like this ‘… yielded similar conclusions (data not shown).’ I cannot find the data. Are there the data elsewhere?

[Discussion]

(9) L311: ‘varied between 0.10 to 0.25’. I cannot understand what the values indicates. I guess that the ‘0.10’ indicates the association between Breastfeeding of ‘6 to <12 months’ and verbal ability at age 11 in Fig.4?? But I cannot find the ‘0.25’ (Maybe ‘0.26’ in the association between breastfeeding of ‘>=12 months’ and verbal at 14?). Most readers cannot identify the places of the values among lots of values in Fig.4 or 5. The same can be said for L366.

(10) L402: Why the authors specifically mention the proportion of missing data only for covariates (n = 662)? Maternal cognitive scores were missed more (n=899) and could yield a risk of selection bias, too.

(11) L374: The authors stated “We also considered the possibility that the remaining associations were explained by residual confounding produced by unmeasured confounders, such as paternal measures of cognitive ability or broader measures of maternal cognitive ability. In addition to them, child’s factors should be considered. If an infant at potential risk of developmental delay has less preference to breastfeeding, a superficial association can be produced between breastfeeding duration and better cognitive abilities that were assessed subsequently.

Reviewer #2: The study finds that SEP approximately halved the effects of breastfeeding on cognitive development and that adjustment for maternal cognitive scores further diluted the effects, but did remove all associations. Although similar results have been observed in other studies, I do find that the quality of the maternal cognitive assessment is an important issue. The specific test applied required the mothers to explain the meaning of several words, and as the authors point out this specific test may no reflect broader intelligence. However, there is not only a validity problem, but also a reliability problem. Thus, it is possible that if a more comprehensive and more reliability test of maternal intelligence had been used, the effects of breastfeeding might have been non-significant after adjustment for maternal cognitive ability.

The authors are aware of this problem, but I think an expanded discussion of the issue would improve the paper.

Reviewer #3: Thank you for the opportunity to review this excellent manuscript by Pereyra-Elias and colleagues. The authors examined the associations between breastfeeding and children’s cognitive development in a large longitudinal cohort study which is nationally representative of the UK. This research question, i.e. does breastfeeding improve intelligence?, is old of near a century, but the field keeps improving with stronger and stronger research design to address it. To fill current gaps, the authors examined more closely i) the extent of confounding in the relationship, in particular by maternal intelligence, which is relatively rarely measured in large studies; and ii) potential long lasting effect up to the adolescence period using repeated cognitive outcomes. The authors found that even after adjusting for socioeconomic factors, and then maternal intelligence, remains an association between longer breastfeeding duration and children’s cognitive development at almost all ages and outcome measures from 5 to 14 y. The paper is also clearly structured and well written. My main comments are as follows:

1) Methods/Study population: lines 73-77 break the flow of the Study population section. It may read better in introduction or discussion or even in the statistical analysis section (when explaining the adjustment procedure).

2) Methods/Exposure: Although the authors cite relevant literature showing that mothers recall well the duration of breastfeeding of their child even at 5 or 6 years postpartum, this retrospective assessment of breastfeeding duration remains one limitation of the present study. On the same matter, the authors argue in the Discussion section that “most of the previous reports have traditionally dichotomised breastfeeding duration as yes/no or with a temporal cut-off [….]. Conversely, our study uses several categories of duration, which helps to explore this relationship in a more nuanced way”. While this is true overall, there are a couple of studies which used breastfeeding duration in an even more nuanced way, by using it as a continuous variable. See for example Tozzi et al. Dev Med Child Neurol 2012, Bernard et al. J Pediatr 2013, Bernard et al. J Pediatr 2017, and Belfort et al. JAMA Pediatr 20213. On top of the retrospective assessment, categorizing the breastfeeding variable is another limitation since it removes information.

3) Methods/Statistical analysis: The selection process of the confounders described on lines 167-170 is pretty contradictory with an a priori approach using DAGs, especially that the authors use a large dataset that does not suffer from a lack of statistical power. I would support forcing variables which are described as confounders in the literature, even when not associated with the outcome in the employed dataset.

4) Methods/Ethical approval: Unless it is among PLoS One requirements, this section would read better right after the Study population section.

5) Results/Breastfeeding duration: using Table 1, I found that 27.7% of participants were never breastfed and 25.7% were breastfed for six months or longer. Not 33.9 and 23.0, respectively. Please explain or correct discrepancies.

6) Results: I am not clear with how the authors consider the interactions between breastfeeding duration and the age at which the cognitive scores were evaluated. In my view, the interaction term is necessary, by default, in all repeated-outcome models. If it is significant, it means that the effect of breastfeeding on the outcome varies by age. If not significant, it means that the effect is quite the same across ages. I do not understand why one would remove the interaction term if not significant.

7) Discussion: The authors seem to sit on the fence regarding the relevance of the effect size which is observed (0.10 to 0.26 SD). They sometimes write it is modest, they sometimes write it should not be underestimated. In my view, it is modest at the clinical level, but it is huge at the population level. I made quick calculations using the presented data (based on the BF prevalence shown in Table 1 and the effect sizes from Model 4 at age 14 y in Fig 4). I transformed SD units into IQ points, because I find it more striking for public health messages. Were all children breastfed 12 months or longer, the average gain (provided the effect is causal) for the overall population would be around 3.2 IQ points. Let’s imagine you and I gain 3.2 IQ points starting tomorrow, it would change very little to our cognitive performance and everyday life. Let’s imagine the whole UK population gains 3.2 IQ points tomorrow, the benefit for the country is hard to believe.

8) Discussion: the authors are quite short on the potential underlying mechanisms of a causal effect of breastfeeding on cognitive development. They are quite a few studies on PUFAs. More development would be welcome. They could also discuss the fact that breastfeeding is compared to non-breastfeeding, i.e., infant formula, whose nutritional composition has largely changed over the last decades. In some way, we usually compare something to another thing that we wrongly consider stable and comparable across study settings. This could explain why findings vary across studies.

No further comments

6. PLOS authors have the option to publish the peer review history of their article (what does this mean?). If published, this will include your full peer review and any attached files.

Reviewer #1: No

Reviewer #2: No

Reviewer #3: **Yes: **Jonathan Y. Bernard

---

## [Author Response · Author response to Decision Letter 0]

5 Apr 2022

Dear editor, 

We would like to thank you and the reviewers for the valuable comments provided. We believe these suggestions have helped us to improve the quality of our manuscript. Please, find below our responses to each of the comments (in Italics):

- We have adapted the manuscript formatting.

2. In your ethics statement in the manuscript and in the online submission form, please provide additional information about the data used in your retrospective study. Specifically, please ensure that you have discussed whether all data were fully anonymized before you accessed them.

- Thank you for this comment. We have clarified that data were pseudonymised (before we accessed it) to prevent participant identification and subsequently uploaded to the UK Data Service. The data we accessed were anonymous to us. This information is in the “Ethical Approval” section of the Methods. 

- We have reviewed our reference list and we have not found any retracted articles. Reference 8, by Simpson DA, et al. has a correction, and we have mentioned the correction in that reference. 

We have also included additional references in the revised version of the manuscript.

Reviewers' comments:

Reviewer #1: This study examined the relationship between breastfeeding duration and cognitive development until children were 14 years old using a longitudinal data of a UK cohort study. Verbal and spatial abilities of the children were repeatedly assessed at 5, 7, 11 and 14 years old. The results indicate that the associations were still positive at 14 years old even after controlling for full confounding factors measured.

Although the manuscript includes a number of valuable data on the beneficial effects of breastfeeding, there were several points that should be clarified.

[Introduction]

(1) L59: (spell error?) ‘ould’ may be ‘could’ or ‘would’.

- We thank Reviewer 1 for spotting this mistake. The word is ‘would’. We have corrected it. 

[Methods]

-Exposure

(2) L111: How the study group collected the data of breastmilk, formula, solid foods, etc. The authors stated ‘Breastfeeding duration was evaluated by maternal responses to the questions: “Did you …”.’. Did they asked the questions when they visited a research centre for child’s assessment or received questionnaires afterwards by mail. The methodologies of the investigation could affect mother’s responses.

- The questions (for all the data) were asked during home interviews conducted by trained personnel using computer-assisted personal interviewing. We have added this information in a new subsection (“Data Collection”) in the Methods section.

-Potential confounders

(3) In DAG (Fig.2), there were a space between ‘Maternal cognitive score’ and ‘Paternal cognitive score’. I first missed the above. The authors may want to remove the space.

- We agree with the reviewer. We have removed the space. 

–Statistical analysis

(4) L156: They used mixed-effects generalised linear models for explaining the repeated measures of cognitive assessments within each child. However, there seem to be no results of overall verbal or spatial ability. Please describe it more clearly.

- In Comment 4, Reviewer 1 requests the inclusion of overall results; while Reviewer 3 in Comment 5, argues that interactions should always be included (which would require us to report separately for each group). Given that the positions of the reviewers seem to differ, we have written a common response for these two comments, explaining our decisions. 

We are not completely sure about what Reviewer 1 means by “overall results”. If they are requesting the mean cognitive scores by breastfeeding duration group and in the overall sample, these can be found in Table 1. If they are requesting the results for all ages combined, we decided not to show the overall results for verbal or spatial abilities because there was evidence of a statistical interaction between breastfeeding duration and age at which the outcome was assessed.

Reviewer 3 also asks about the inclusion of interaction terms in the models, whereby the effect of breastfeeding on cognitive scores would differ by the age at which the outcome was assessed. We have included interaction terms between breastfeeding and age of outcome assessment in all of our models and all these interaction terms were statistically significant. This means that there is evidence that the effect of breastfeeding duration on cognitive outcomes differed according to the age at which the outcome was measured. We have kept all the interaction terms in the models, and we have therefore presented parameters for each age separately (5, 7, 11 and 14) and no overall estimates. 

(5) L163: “Models 1 and 2 adjusted for a basic set of confounders (maternal age, ethnicity and language spoken at home) and SEP markers, respectively.” This could mean that model 1 adjusted for a basic set of confounders but not for SEP while model 2 adjusted for SEP but not for the basic set. However, actually, in the legends of Fig.4 and 5, “Model 2 adjusted for Model 1 + SEP”. Please describe them consistently.

- We agree with Reviewer 1. We have rewritten this description. 

[Results]

(6) In Table 1, ‘§’ may be missed at ‘Gestational age at birth’; ‘Maternal age (years)’; the four items of ‘Spatial cognitive scores’, in the ‘Characteristics’ column.

- We added the symbol to those variables. Thanks. 

(7) L231-234: The authers may want to write like this, “… and the language spoken in the household (model 1), … Further adjustment for maternal cognitive scores (model 4) attenuated …”. The descriptions are a bit hart to follow as it was.

- We have specified the model in the description of these results. 

(8) L322: The authors may write like this ‘… yielded similar conclusions (data not shown).’ I cannot find the data. Are there the data elsewhere?

- We have included these results as supplementary material (S6 to S9). 

[Discussion]

(9) L311: ‘varied between 0.10 to 0.25’. I cannot understand what the values indicates. I guess that the ‘0.10’ indicates the association between Breastfeeding of ‘6 to <12 months’ and verbal ability at age 11 in Fig.4?? But I cannot find the ‘0.25’ (Maybe ‘0.26’ in the association between breastfeeding of ‘>=12 months’ and verbal at 14?). Most readers cannot identify the places of the values among lots of values in Fig.4 or 5. The same can be said for L366.

- We thank the reviewer for the comment. We have clarified both instances in the revised text now reads: 

“The fully-adjusted coefficients where there is evidence of an effect of breastfeeding on verbal cognitive scores varied between 0.08 (age 7; <2 months vs never breastfed) to 0.26 SD (age 14; ≥12 months vs never breastfed). For spatial scores, the coefficients varied between 0.08 (age 7; <2 months vs never breastfed) to 0.19 SD (ages 7 and 11; 4 to <6 months vs never breastfed).”. 

(10) L402: Why the authors specifically mention the proportion of missing data only for covariates (n = 662)? Maternal cognitive scores were missed more (n=899) and could yield a risk of selection bias, too.

- We corrected the statement (second paragraph of Limitations subsection). We have included the proportion of data missing for all covariates and for maternal cognitive scores. We have also adapted the Flowchart (Figure 3) to show the number of participants lost for missing data in a sequential manner. 

(11) L374: The authors stated “We also considered the possibility that the remaining associations were explained by residual confounding produced by unmeasured confounders, such as paternal measures of cognitive ability or broader measures of maternal cognitive ability.” In addition to them, child’s factors should be considered. If an infant at potential risk of developmental delay has less preference to breastfeeding, a superficial association can be produced between breastfeeding duration and better cognitive abilities that were assessed subsequently.

- We agree with Reviewer 1. To avoid introducing confounding by child’s developmental impairment, we have excluded those children who are most likely to have those conditions, i.e. participants born premature. 

Reviewer #2: The study finds that SEP approximately halved the effects of breastfeeding on cognitive development and that adjustment for maternal cognitive scores further diluted the effects, but did remove all associations. Although similar results have been observed in other studies, I do find that the quality of the maternal cognitive assessment is an important issue. The specific test applied required the mothers to explain the meaning of several words, and as the authors point out this specific test may no reflect broader intelligence. However, there is not only a validity problem, but also a reliability problem. Thus, it is possible that if a more comprehensive and more reliability test of maternal intelligence had been used, the effects of breastfeeding might have been non-significant after adjustment for maternal cognitive ability.

The authors are aware of this problem, but I think an expanded discussion of the issue would improve the paper.

- We would like to thank Reviewer 2 for the comments. We have included more detail on the test used to measure maternal cognitive ability (Methods, “Potential confounders” subsection). We have also expanded the discussion on how the different maternal cognitive measures could affect the final estimates (first paragraph of the Limitations section). 

Reviewer #3: Thank you for the opportunity to review this excellent manuscript by Pereyra-Elias and colleagues. The authors examined the associations between breastfeeding and children’s cognitive development in a large longitudinal cohort study which is nationally representative of the UK. This research question, i.e. does breastfeeding improve intelligence?, is old of near a century, but the field keeps improving with stronger and stronger research design to address it. To fill current gaps, the authors examined more closely i) the extent of confounding in the relationship, in particular by maternal intelligence, which is relatively rarely measured in large studies; and ii) potential long lasting effect up to the adolescence period using repeated cognitive outcomes. The authors found that even after adjusting for socioeconomic factors, and then maternal intelligence, remains an association between longer breastfeeding duration and children’s cognitive development at almost all ages and outcome measures from 5 to 14 y. The paper is also clearly structured and well written. 

My main comments are as follows:

1) Methods/Study population: lines 73-77 break the flow of the Study population section. It may read better in introduction or discussion or even in the statistical analysis section (when explaining the adjustment procedure).

- We thank the reviewer for the suggestion. These lines include the rationale behind some selection criteria of our sample. Therefore, we have decided to leave the paragraph as it currently is. The other two reviewers have not pointed this out, but if the Editor feels strongly about this, we could change it. 

2) Methods/Exposure: Although the authors cite relevant literature showing that mothers recall well the duration of breastfeeding of their child even at 5 or 6 years postpartum, this retrospective assessment of breastfeeding duration remains one limitation of the present study. On the same matter, the authors argue in the Discussion section that “most of the previous reports have traditionally dichotomised breastfeeding duration as yes/no or with a temporal cut-off [….]. Conversely, our study uses several categories of duration, which helps to explore this relationship in a more nuanced way”. While this is true overall, there are a couple of studies which used breastfeeding duration in an even more nuanced way, by using it as a continuous variable. See for example Tozzi et al. Dev Med Child Neurol 2012, Bernard et al. J Pediatr 2013, Bernard et al. J Pediatr 2017, and Belfort et al. JAMA Pediatr 20213. On top of the retrospective assessment, categorizing the breastfeeding variable is another limitation since it removes information.

- We agree with Reviewer 3. The retrospective nature of the exposure could carry information bias, however, we consider that this is minimal, as breastfeeding duration was evaluated at the age of 9 months for the majority of the cohort. It was only asked again at the age of 5 years, for only less than 10% of participants who were breastfed for longer than 9 months. We have added this information in the Methods section. 

The categorisation of this variable is likely to be less informative than its continuous form. However, we decided to categorise it because, as it can be seen in Figures 4 and 5, not all the associations are linear (this is especially true for spatial scores). We consider that the categorisation of breastfeeding duration facilitates the interpretation of the results for a wider audience. The categorisation is also fine enough to provide information about potential patterns in the association, such as dose-response relationships. 

We would also like to thank the Reviewer for suggesting the studies from Tozzi et al. and Bernard et al. We have added them to our references. 

3) Methods/Statistical analysis: The selection process of the confounders described on lines 167-170 is pretty contradictory with an a priori approach using DAGs, especially that the authors use a large dataset that does not suffer from a lack of statistical power. I would support forcing variables which are described as confounders in the literature, even when not associated with the outcome in the employed dataset.

- We understand the concerns of the reviewer. We followed a mixed approach for variable selection, based both on epidemiological and statistical criteria. The inclusion of our main confounders (markers of socioeconomic position and maternal cognitive scores) and other important confounding variables (such as gestational age at birth, ethnicity and language) was decided a priori. While we have a considerably large sample size, we also had a comprehensive list of variables available to us that could behave as potential confounders according to the literature. Therefore, in order to select a more parsimonious final model that would control for the effect of these potentially confounding factors, we decided to use statistical criteria for the selection of these variables. 

4) Methods/Ethical approval: Unless it is among PLoS One requirements, this section would read better right after the Study population section.

- We agree with Reviewer 3. We have relocated the section “Ethical approval”. 

5) Results/Breastfeeding duration: using Table 1, I found that 27.7% of participants were never breastfed and 25.7% were breastfed for six months or longer. Not 33.9 and 23.0, respectively. Please explain or correct discrepancies.

- These discrepancies exist because the 27.7% calculated from Table 1 (2,179/7,855) does not consider the complex sample design. Design and non-response weights developed by the MCS team were used to account for the complex sampling strategy and the impact of attrition on the estimates of prevalence and the standard errors (and therefore confidence intervals) for the measures of effect. The correct (weighted) percentages are the “33.9%” and “23.0%” found in the text. This is explained in the Methods and also in the Table 1’s footnotes.

6) Results: I am not clear with how the authors consider the interactions between breastfeeding duration and the age at which the cognitive scores were evaluated. In my view, the interaction term is necessary, by default, in all repeated-outcome models. If it is significant, it means that the effect of breastfeeding on the outcome varies by age. If not significant, it means that the effect is quite the same across ages. I do not understand why one would remove the interaction term if not significant.

- In Comment 4, Reviewer 1 requests the inclusion of overall results; while Reviewer 3 in Comment 5, argues that interactions should always be included (which would require us to report separately for each group). Given that the positions of the reviewers seem to differ, we have written a common response for these two comments, explaining our decisions. 

We are not completely sure about what Reviewer 1 means by “overall results”. If they are requesting the mean cognitive scores by breastfeeding duration group and in the overall sample, these can be found in Table 1. If they are requesting the results for all ages combined, we decided not to show the overall results for verbal or spatial abilities because there was evidence of a statistical interaction between breastfeeding duration and age at which the outcome was assessed.

Reviewer 3 also asks about the inclusion of interaction terms in the models, whereby the effect of breastfeeding on cognitive scores would differ by the age at which the outcome was assessed. We have included interaction terms between breastfeeding and age of outcome assessment in all of our models and all these interaction terms were statistically significant. This means that there is evidence that the effect of breastfeeding duration on cognitive outcomes differed according to the age at which the outcome was measured. We have kept all the interaction terms in the models, and we have therefore presented parameters for each age separately (5, 7, 11 and 14) and no overall estimates. 

7) Discussion: The authors seem to sit on the fence regarding the relevance of the effect size which is observed (0.10 to 0.26 SD). They sometimes write it is modest, they sometimes write it should not be underestimated. In my view, it is modest at the clinical level, but it is huge at the population level. I made quick calculations using the presented data (based on the BF prevalence shown in Table 1 and the effect sizes from Model 4 at age 14 y in Fig 4). I transformed SD units into IQ points, because I find it more striking for public health messages. Were all children breastfed 12 months or longer, the average gain (provided the effect is causal) for the overall population would be around 3.2 IQ points. Let’s imagine you and I gain 3.2 IQ points starting tomorrow, it would change very little to our cognitive performance and everyday life. Let’s imagine the whole UK population gains 3.2 IQ points tomorrow, the benefit for the country is hard to believe.

- We believe this is a fair point. An IQ gain of 3.2 points could have important at the population level. We have added a sentence in the Conclusions section. 

We also believe that there is reasonable uncertainty regarding the true size of this estimate. As we point out, there might be residual confounding (for example, the effect of paternal intelligence). While we believe that it is unlikely that the effect would be completely explained after further controlling for this residual confounding (as suggested by the E-values), the final effect sizes would probably be smaller than the current estimates (which range from 0.08 to 0.26). In that sense, while we believe that there is an effect, this is most likely modest. Therefore, we preferred to be cautious in our interpretation of the results in the Discussion section and throughout the paper.

8) Discussion: the authors are quite short on the potential underlying mechanisms of a causal effect of breastfeeding on cognitive development. They are quite a few studies on PUFAs. More development would be welcome. They could also discuss the fact that breastfeeding is compared to non-breastfeeding, i.e., infant formula, whose nutritional composition has largely changed over the last decades. In some way, we usually compare something to another thing that we wrongly consider stable and comparable across study settings. This could explain why findings vary across studies.

- We thank the reviewer for these comments. We have expanded on potential underlying mechanisms that explain the association between breastfeeding and cognitive development. We also discussed comparability between study settings. Please see the paragraph that precedes the Limitations subsection. 

Thanks again for the comments. 

Kind regards, 

Reneé Pereyra-Elías

On behalf of the authors

---

## [Editor Report · Decision Letter 1]

7 Apr 2022

To what extent does confounding explain the association between breastfeeding duration and cognitive development up to age 14?  Findings from the UK Millennium Cohort Study

PONE-D-22-03469R1

Dear Dr. Pereyra-Elías,

We’re pleased to inform you that your manuscript has been judged scientifically suitable for publication and will be formally accepted for publication once it meets all outstanding technical requirements.

Kind regards,

Emma K. Kalk

Academic Editor

PLOS ONE
---

## [Editor Report · Acceptance letter]

3 May 2022

PONE-D-22-03469R1 

To what extent does confounding explain the association between breastfeeding duration and cognitive development up to age 14?  Findings from the UK Millennium Cohort Study 

Dear Dr. Pereyra-Elías:

I'm pleased to inform you that your manuscript has been deemed suitable for publication in PLOS ONE. Congratulations! Your manuscript is now with our production department. 

Kind regards, 

on behalf of

Dr. Emma K. Kalk 

Academic Editor

PLOS ONE